# Anatomical Study of the Lateral Tibial Spine as a Landmark for Weight Bearing Line Assessment during High Tibial Osteotomy

**DOI:** 10.3390/medicina59091571

**Published:** 2023-08-29

**Authors:** Tae Woo Kim, June Seok Won

**Affiliations:** Department of Orthopedic Surgery, Seoul National University College of Medicine, Seoul National University Boramae Medical Center, Seoul 07061, Republic of Korea; chris9906@naver.com

**Keywords:** high tibial osteotomy, planning, landmark, lateral tibial spine, anatomy

## Abstract

*Background:* Accurate pre-operative planning is essential for successful high tibial osteotomy (HTO). The lateral tibial spine is a commonly used anatomical landmark for weight-bearing line assessment. However, studies on the mediolateral (M-L) position of the lateral tibial spine on the tibial plateau and its variability are limited. *Purpose:* This study aimed to (1) analyze the M-L position of the lateral tibial spine on the tibial plateau and its variability, (2) investigate radiologic parameters that affect the position of the lateral tibial spine, and (3) determine whether the lateral tibial spine can be a useful anatomical landmark for weight-bearing line assessment during HTO. *Materials and Methods:* Radiological evaluation was performed on 200 participants (64% female, mean age 42.3 ± 13.2 years) who had standing anterior–posterior plain knee radiographs with a patellar facing forward orientation. The distances from the medial border of the tibial plateau to the lateral spine peak (dLSP) and lateral spine inflection point (dLSI) were measured using a picture archiving and communication system. The medial–lateral inter-spine distance (dISP) was also measured. All parameters were presented as percentages of the entire tibial plateau width. The relationships between the parameters were also investigated. *Results:* The mean value of dLSP was 56.9 ± 2.5 (52.4–64.5)%, which was 5% lower than the Fujisawa point (62%). The mean value of dLSI was 67.9 ± 2.2 (63.4–75.8)%, which was approximately 5% higher than the Fujisawa point. The values of the dLSP and dLSI were variable among patients, and the upper and lower 10% groups showed significantly higher and lower dLSP and dLSI, respectively, than the middle 10% group. The mean value of dISP was 16.5 ± 2.4%, and it was positively correlated with dLSP and dLSI. *Conclusions:* On average, the dLSP and dLSI were located −5% and +5% laterally from the conventional Fujisawa point, and they may be useful landmarks for correction amount adjustment during HTO. However, it should be noted that correction based on the lateral tibial spine can be affected by anatomical variations, especially in patients with small or large inter-spine distances.

## 1. Introduction

High tibial osteotomy (HTO) is an efficient surgical option for treating middle-aged patients with medial knee osteoarthritis (OA) and varus alignment [1,2,3,4,5]. Accurate alignment correction is essential for successful HTO in patients with OA [6,7,8,9,10,11]. Under-correction can result in residual knee pain and shorten the longevity of the effects of this procedure [12]. Over-correction can induce lateral and patellofemoral arthritis and cause dissatisfaction associated with excessive valgus deformity [13,14,15].

To reduce the correction error during HTO, pre-operative planning based on weight-bearing line assessment plays an important role, and the first step in HTO pre-operative planning is the determination of the appropriate target point on the tibial plateau [16,17,18,19,20,21]. Since it was first documented in 1979, the Fujisawa point (62–63%) has been commonly used as an ideal realignment target during HTO, and the outcome of HTO has also been evaluated using this reference point [22,23,24]. However, some authors recommend adjusting the correction target depending on the severity of osteoarthritis and the reason for surgery [2,25].

The lateral tibial spine can also be used as an anatomical landmark for weight-bearing line assessment during HTO. Lee et al. reported that medial opening-wedge HTOs aimed at the lateral tibial spine showed comparable clinical outcomes to those of HTOs targeting the conventional Fujisawa point [26]. The lateral tibial spine is a bony eminence on the tibial plateau to which the anterior cruciate ligament attaches, which varies anatomically between individuals [27,28]. However, it is unclear whether the lateral tibial spine can be a constant anatomical landmark for HTO in the mediolateral dimension of the tibial plateau.

Therefore, this study aimed to (1) analyze the M-L position of the lateral tibial spine on the tibial plateau and its variability, (2) investigate the radiological parameters that affect the position of the lateral tibial spine, and (3) determine whether the lateral tibial spine can be a useful anatomical landmark for weight-bearing line assessment during HTO. We hypothesized that the lateral tibial spine can be a useful anatomical landmark for HTO and that it can also show anatomical variation correlated with specific radiologic parameters.

## 2. Materials and Methods

### 2.1. Study Population

A total of 517 consecutive standing knee anteroposterior (AP) radiographs of 517 patients diagnosed with medial compartment knee osteoarthritis at Seoul National University Boramae Medical Center between January 2019 and March 2020 were retrospectively reviewed. Of these, 317 radiographs were excluded, and the remaining 200 knee AP radiographs were examined. The exclusion criteria were as follows: (1) Kellgren–Lawrence grade IV with a large osteophyte; (2) bone deformity of the knee due to previous surgery or trauma; (3) rheumatoid arthritis; (4) infection; and (5) absence of a standing knee AP radiograph with proper lower extremity rotation (i.e., with the patella facing forward). In patients with bilateral knee OA, only the side with the more progressed knee OA was evaluated. Patient demographic data, including age and sex, and data on laterality were also collected. This study was approved by the Institutional Review Board (IRB No.20-2021-22), and the requirement for informed consent was waived due to the retrospective study design.

### 2.2. Radiologic Assessment

Standing knee AP radiographs were obtained using a UT 2000 X-ray machine (Philips Research, Eindhoven, The Netherlands) set at 90 Kv and 50 mA. Knee rotation was controlled using our standard protocol to locate the patella at the center of the femoral condyle during the examination. All radiographs were digitally acquired using a picture archiving and communication system (PACS), and radiographic assessments were performed using the PACS (INFINITT, Seoul, Republic of Korea). The standard knee AP view was marked with a digital ruler, and each value was calibrated automatically on the PACS system.

The peak and inflection points of the lateral tibial spine were determined as the most prominent points and the intersection point between the tibial spine slope and the tibial plateau, respectively. The distance between the medial tibial border and lateral spine peak point (dLSP), the distance between the medial tibial border and lateral spine inflection point (dLSI), and the mediolateral inter-spine distance (dISP) were measured and presented as a percentage of the entire tibial plateau width (Figure 1).

For the interobserver reliability test, two orthopedic surgeons independently measured the radiographic parameters. Each observer was blinded to the measurements of the other observer. For the intra-observer reliability test, each observer measured the radiological parameters twice at 8-week intervals.

### 2.3. Analysis of Radiologic Parameters

To quantitatively analyze the location of the lateral tibial spine on the tibial plateau, the dLSP and dLSI were measured. To evaluate the variability in each parameter statistically, the upper 10% and lower 10% groups were compared with the middle 10% group, respectively. To determine whether other radiologic parameters could predict the amount of dLSP or dLSI, correlations between dLSP, dLSI, tibial plateau M-L width, and dISP were analyzed.

### 2.4. Statistical Analysis

All statistical analyses were performed using SPSS version 25.0 (IBM, Armonk, NY, USA). An a priori power analysis was performed with the assumption that a 2.0% M-L dimension was radiologically significant, with a standard deviation of 2.4% mm. It was found that 20 knees in each group were sufficient for all parameters (*α*  =  0.05, *β*  =  0.8). *p*-values of <0.05 were considered statistically significant. All data are presented as mean and standard deviation. Student’s *t*-test was used for intergroup comparisons. Correlation analyses between radiological parameters were performed using Pearson’s correlation coefficients. Inter- and intra-observer reliabilities were evaluated using the intra-class correlation coefficient (ICC).

## 3. Results

In total, 200 standing AP knee radiographs (115 right and 85 left) from 200 patients (92 males, 108 females) were evaluated. The mean age of the patients was 56.2 ± 7.2 years, and the severity of knee OA was K-L grades I, II, and III in 14, 76, and 110 patients, respectively (Table 1). For all radiological parameters, the inter- and intra-observer ICC were >0.8 (Table 2).

In the quantitative analysis of the lateral tibial spine position, the mean values of dLSP, dLSI, and dISP were 56.9 ± 2.5%, 67.9 ± 2.2%, and 16.5 ± 2.4%, respectively. The peak point and inflection point of the lateral tibial spine were located approximately 5% lower and 5% higher, respectively, than the conventional Fujisawa point (62–63%). However, both dLSP and dLSI varied broadly between individuals. The maximum and minimum values of dLSP were 64.5% and 52.4%, respectively, and the maximum and minimum values of dLSI were 75.8% and 63.4%, respectively. The upper 10% group showed significantly higher dLSP and dLSI than the middle 10% group (*p* = 0.001, *p* = 0.002); the lower 10% group also showed significantly lower dLSP and dLSI than the middle 10% group (*p* = 0.001, *p* = 0.001) (Figure 2 and Figure 3).

In the correlation analysis, the dISP showed significant positive correlations with the dLSP (r = 0.787, *p* = 0.001) and dLSI (r = 0.756, *p* = 0.001) (Figure 4). In patients with larger inter-spine distances, the peak and inflection points of the lateral tibial spine were located more laterally. Similarly, in patients with a narrow inter-spine distance, the peak and inflection points of the lateral tibial spine were located more medially. However, the tibial plateau M-L width, sex, and laterality did not affect the dLSP or dLSI (*p* > 0.05) (Table 3).

The mean value of dLSP was 56.9 ± 2.5%, and the maximum and minimum values of dLSP were 64.5% and 52.4%, respectively. The upper 10% and lower 10% groups showed significantly higher and lower dLSP than the middle 10% group, respectively.

The mean value of dLSI was 67.9 ± 2.2%, and the maximum and minimum values of dLSI were 75.8% and 63.4%, respectively. The upper 10% and lower 10% groups showed significantly higher and lower dLSP than the middle 10% group, respectively.

In the correlation analysis, dISP showed a significantly positive correlation with dLSP (r = 0.787, *p* < 0.05) and dLSI (r = 0.756, *p* < 0.05). In patients with large and small dISP, dLSP and dLSI also showed increasing and decreasing tendency, respectively.

## 4. Discussion

The principal finding of this study was that the lateral tibial spine could be a useful anatomical landmark for weight-bearing line assessment during HTO. The mean values of the peak and inflection points of the lateral tibial spine were located approximately −5% and +5% laterally from the conventional Fujisawa point, and it could be easily used for adjustment of the amount of correction. However, the values of the dLSP and dLSI varied among patients, and dISP correlated with dLSP and dLSI. Therefore, the hypothesis of this study was fully supported.

Previous studies on HTO that used the lateral tibial spine as a landmark for alignment correction reported results similar to those of the present study. Lee et al.’s study that compared HTOs aiming at either the peak of the lateral tibial spine or the Fujisawa point showed that the lateral spine peak point was located at 57.4 ± 1.6% from the medial border of the tibia [26]. Jiang et al.’s study also showed that the mean WBL percentage of the top of the tibial spine was 57.7 ± 2.1% [29]. However, the inflection point of the lateral tibial spine in our study was quite different from that of Jiang et al. (67.9 ± 2.2% vs. 74.6 ± 3.3%). In this study, the lateral end of the lateral tibial spine was defined as the inflection point where the tibial spine slope and tibial plateau intersect. On the other hand, in Jiang et al.’s study, the bottom point of the lateral slope was defined as the intersection point between the lateral slope of the lateral tibial spine and the joint line of the tibia. Since the lateral tibial slope and tibial plateau intersect in a round shape rather than an acute angle, the bottom point is located further to the lateral side than the inflection point. This discrepancy can be attributed to the different measurement methods of the intersection point between the tibial spine slope and the tibial plateau.

Quantitative analysis of the lateral tibial spine can be helpful in adjusting the amount of correction during HTO. Jakob et al. suggested that correction of the mechanical axis should differ based on the thickness of cartilage loss in the medial compartment (1/3 cartilage loss: WBL 55–57.5%, 2/3 cartilage loss: WBL 60–62.5%, full thickness cartilage loss: WBL 65–67.5%) [25]. Based on our results, the location of the peak, mid, and inflection points of the lateral tibial spine correspond well with Jakob et al.’s suggestion, depending on different arthroses of the medial compartment. Recent studies have also shown that HTO targeting the peak point of the lateral tibial spine showed clinical outcomes comparable to those of HTO targeting the conventional Fujisawa point [26]. Adjustment of the correction angle may also be necessary depending on the purpose of HTO. When anterior cruciate ligament (ACL) or posterior cruciate ligament (PCL) reconstruction fails in patients with varus deformity, neutral or slight varus alignment is usually targeted during HTO. Even during protective realignment surgery combined with cartilage regeneration procedures, the WBL target is a little more medial than the conventional Fujisawa point. In these cases, the peak point of the lateral tibial spine can be a more useful anatomical landmark than the conventional Fujisawa point.

Another remarkable finding of our study was that the dLSP and dLSI varied among patients. The discrepancy between maximum and minimum values in dLSP, and dLSI were more than 10%, respectively. Also, the upper 10% group showed significantly higher dLSP and dLSI than the middle 10% group; the lower 10% group also showed significantly lower dLSP and dLSI than the middle 10% group. These results can be explained by anatomical variation in the location and size of the lateral tibial spine. Anatomical studies that indicate variable tibial footprint sizes of the ACL among individuals may also indirectly support the variability in the inter-spine distance [27,28].

In the correlation analysis, dLSP and dLSI were significantly correlated with inter-spine distance in this study. Although the dLSP and dLSI were located −5%, and +5% laterally from the conventional Fujisawa point on average, anatomical variation can be an obstacle that reduces the accuracy and efficiency of dLSP or dLSI as an anatomical landmark for HTO. The inter-spine distance can be easily used to predict large or small dLSP or dLSI and to avoid over- or under-correction without direct measurement of each point. To the best of our knowledge, this is the first study to document anatomical variation in the lateral tibial spine and the radiologic predictor for anatomical variance in the lateral tibial spine.

This study had some limitations. First, this study investigated osteoarthritic knees in an Asian population, and the results may be different from those of a Western population. However, the location of the peak and inflection points in the lateral tibial spine was presented as a percentage of the entire tibial plateau width in this study, which may remove the effect of individual size differences in the tibial plateau and tibial spine. Further anatomical studies in different races are necessary to generalize the results of this study. Second, in contrast to the peak point method, the method used to define the intersection point between the lateral tibial spine and the tibial plateau can differ between examiners. The different values of the intersection point in the lateral tibial spine and tibial plateau between Jiang et al.’s study (bottom point) and this study (inflection point) may be related to different measurement methods, and more effort to reach a consensus is required. Third, in this study, the correlation between dLSP and dLSI and diverse radiologic parameters was not investigated. The femoral notch width, which is known to be related to the size of the ACL footprint, can also be a predictor for dLSP and dLSI. Thus, it is an interesting topic for future study. Fourth, this study was conducted using a standing knee AP image, which may be different from the actual situation in which HTO planning is performed in the long leg view. However, in our hospital, a patient’s posture is identical between the standing knee AP view and the long leg view, and we believe that radiologic parameters related to the lateral tibial spine may not be different between the standing knee AP view and the long leg view.

## 5. Conclusions

On average, the dLSP and dLSI were located −5%, and +5% laterally from the conventional Fujisawa point, and they may be useful landmarks for correction amount adjustment during HTO. However, it should be noted that correction based on the lateral tibial spine can be affected by anatomical variations, especially in patients with small or large inter-spine distances.

## Figures and Tables

**Figure 1 medicina-59-01571-f001:**
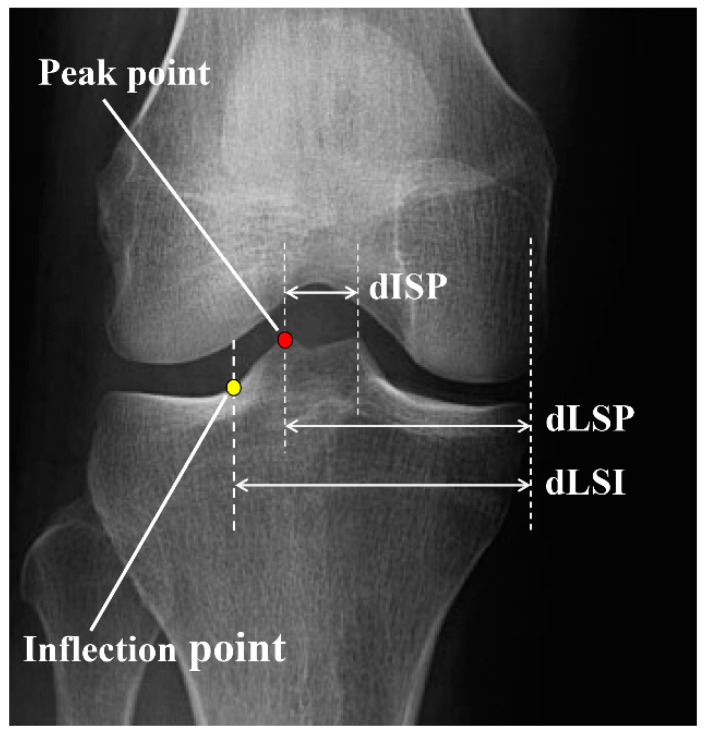
Radiographic measurements of the lateral tibial spine. Peak point: the most prominent point of the lateral tibial spine, inflection point: the intersection point between the tibial spine slope and tibial plateau, dLSP: distance between the tibial medial border and lateral spine peak point, dLSI: distance between the tibial medial border and lateral spine inflection point; dISP: medio-lateral inter-spine distance.

**Figure 2 medicina-59-01571-f002:**
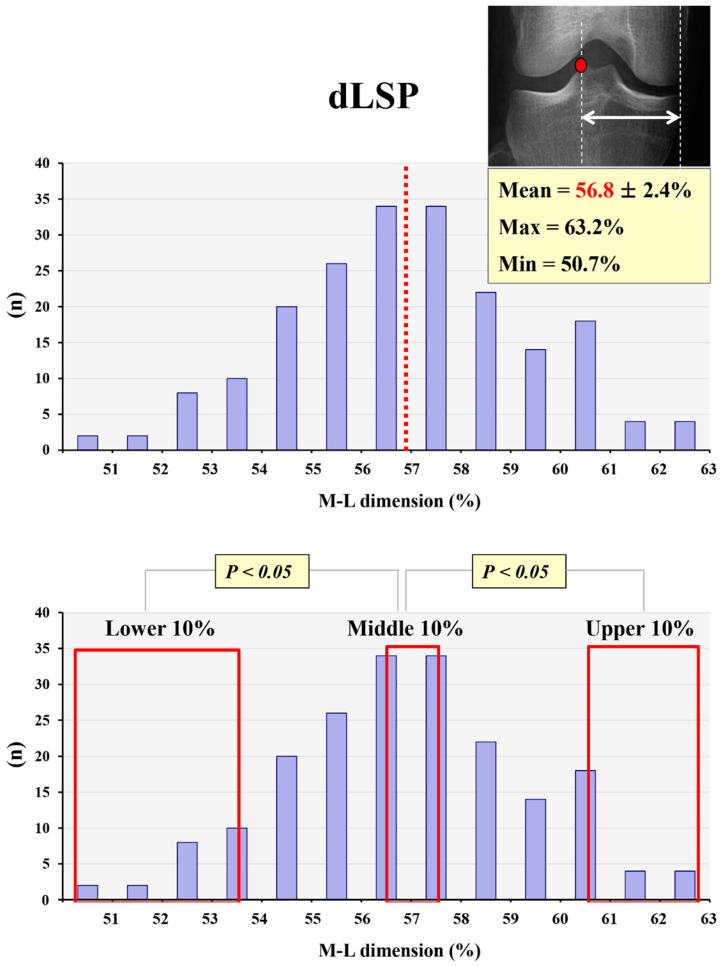
The mean and maximum/minimum values for dLSP and its variation. Red dot: peak point of lateral tibial spine, red dotted line and red font: mean value of dLSP, red frame: lower, middle, and upper 10% groups.

**Figure 3 medicina-59-01571-f003:**
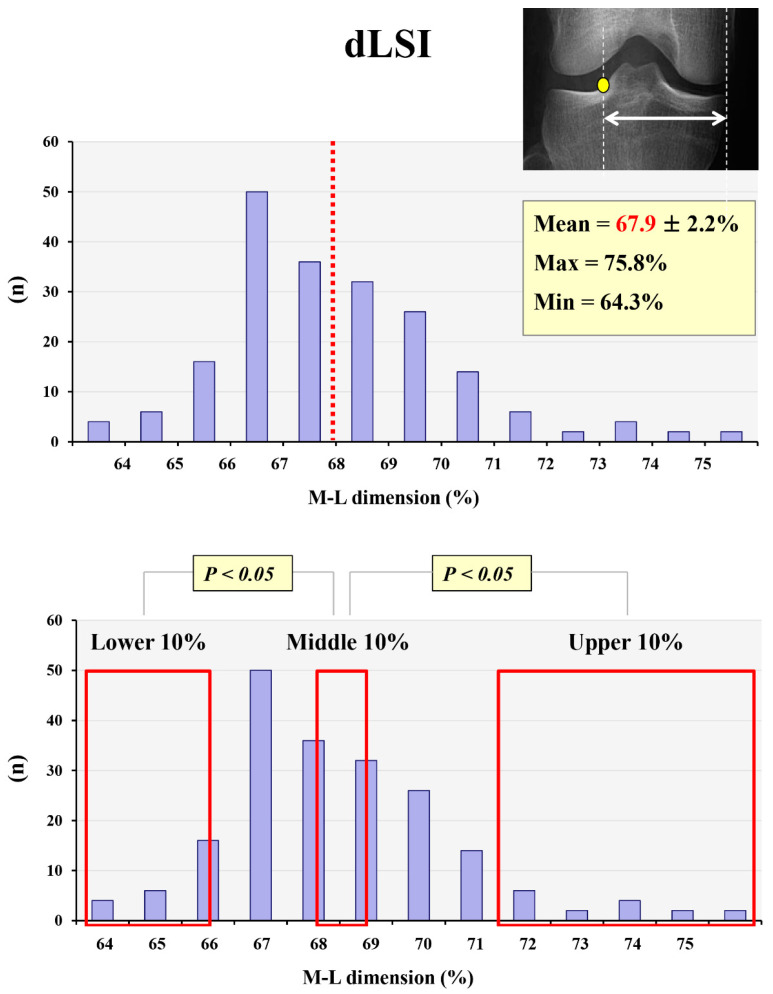
The mean and maximum/minimum values of dLSI and its variation. Yellow dot: inflection point of lateral tibial spine, red dotted line and red font: mean value of dLSI, red frame: lower, middle, and upper 10% groups.

**Figure 4 medicina-59-01571-f004:**
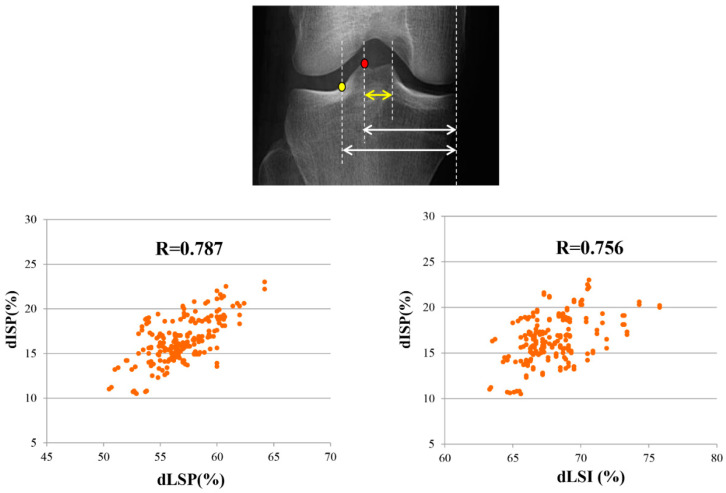
Correlation analysis between radiologic parameters. Yellow dot: inflection point of lateral tibial spine, red dot: peak point of lateral tibial spine, yellow arrow: dISP, shorter white arrow: dLSP, longer white arrow: dLSI.

**Table 1 medicina-59-01571-t001:** Patient demographics and measurements of radiologic parameters.

	Mean	SD (Min–Max)
Patient demographics		
age	56.2	7.2 (40–72)
gender	male (46%)	
laterality	right (57.5%)	
K-L grade	grade I: 14 grade II: 76 grade III: 110	
Radiologic parameters		
Tibial width (mm)	65.3	3.4 (60.1–73.4)
dLSP (%)	56.9	2.5 (52.4–64.5)
dLSI (%)	67.9	2.2 (63.4–75.8)
dISP (%)	16.5	2.4 (12.4–23.1)

K-L, Kellgren–Lawrence; dLSP, the distance between the tibial medial border and lateral spine peak point; dLSI, the distance between the tibial medial border and lateral spine inflection point; dISP, medio-lateral inter-spine distance.

**Table 2 medicina-59-01571-t002:** Intra-observer and inter-observer reliability of radiographic measurements.

	Intra-Observer Reliability	Inter-Observer Reliability
Tibial width (mm)	0.971 (0.966–0.979)	0.936 (0.946–0.952)
dLSP	0.995 (0.991–0.997)	0.996 (0.992–0.998)
dLSI	0.981 (0.966–0.989)	0.991 (0.984–0.995)
dISP	0.988 (0.980–0.993)	0.997 (0.995–0.998)

dLSP, the distance between the tibial medial border and lateral spine peak point; dLSI, the distance between the tibial medial border and lateral spine inflection point; dISP, medio-lateral inter-spine distance.

**Table 3 medicina-59-01571-t003:** Comparison of radiologic parameters depending on gender and laterality.

	dLSI (%)	dLSP (%)	dISP (%)
Male	68.0 ± 2.2	57.0 ± 2.7	16.7 ± 2.3
Female	67.7 ± 2.3	56.8 ± 2.5	16.4 ± 2.3
*p*-value	0.352	0.512	0.241
Right	67.9 ± 2.1	56.7 ± 2.3	16.5 ± 2.2
Left	67.8 ± 2.2	56.8 ± 2.1	16.6 ± 2.1
*p*-value	0.671	0.490	0.540

## Data Availability

The data presented in this study are available on request from the corresponding author. The data are not publicly available due to privacy.

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
