# Peer review of "Anatomical Study of the Lateral Tibial Spine as a Landmark for Weight Bearing Line Assessment during High Tibial Osteotomy"

_medicina, 2023, doi:10.3390/medicina59091571_

Round 1

Reviewer 1 Report

This is an interesting study about preop planning for HTO. The authors have done a valiant effort studing a big number of population. However there are quite a few methodological flows.

Most importantly, when planning for any osteotomy of the knee, one has to aquire calibrated long leg views. Why was that not done? Is a standard AP view enough to provide accurate measurements? Even so, were these xrays calibrated? Are the real sizes measured here?

Secondly, in the very first xray of the paper the lines are in wrong places, compared to the following xrays. This needs correction. The points measured should be specific anatomical landmarks and should be accurately described in the methods section. The authors have done that but they detect a discrepancy with another study in the discussion section. 

I found it very difficult to follow the flow of the paper. The authors have devided their measurements in upper ,lower, middle 10% . What is the reason for that? I personally found that confusing to read.

Was there really no statistical significant differences between genders? It would be useful to see that in a table or figure.

Results such as "larger inter spine distance equaled lateralised spine and vise versa" refer to a more general knee anatomy and is corellated with the size of the femoral notch (stenotic,type A or W) .It would be interesting to see that in the future.

Last but not least serious English grammar and spelling review is required.

Line 27 : can be affected

Line 56: posterior

Line 90 : analyze

Line 108: KL II

Full English gramar review please. Shorter sentences to make the reading easier. 

Author Response

This is revision of " Analysis of lateral tibial spine as an atomical landmark for weight
bearing line assessment in high tibial osteotomy" with article number “medicina-2544355”. Thank you very much for your detailed comments and suggestions to make this a better paper. We have changed our article as the reviewers have suggested,and highlighted the changes in bold.

Reviewer 1

This is an interesting study about preop planning for HTO. The authors have done a valiant effort studing a big number of population. However there are quite a few methodological flows.

  1. Most importantly, when planning for any osteotomy of the knee, one has to aquire calibrated long leg views. Why was that not done? Is a standard AP view enough to provide accurate measurements? Even so, were these xrays calibrated? Are the real sizes measured here?

RESPONSE:

Thank you for your valuable comment. In this study, radiologic parameters were measured on knee standard AP view marked with digital ruler, and each value were calibrated automatically on the PACS system. We additionally described the methods of radiologic measurements in more detail at “2.2 Radiologic Assessment”

  1. Secondly, in the very first xray of the paper the lines are in wrong places, compared to the following xrays. This needs correction. The points measured should be specific anatomical landmarks and should be accurately described in the methods section. The authors have done that but they detect a discrepancy with another study in the discussion section. 

RESPONSE:

Thank you for your valuable comment. We corrected figure 1 as your comment. Anatomical landmarks that define radiologic parameters were described in the method section. The discrepancy in definition of radiologic parameters between this study and another study was described additionally at “Discussion”.

  1. I found it very difficult to follow the flow of the paper. The authors have devided their measurements in upper ,lower, middle 10% . What is the reason for that? I personally found that confusing to read.

RESPONSE:

Thank you for your valuable comment. In this study, upper and lower 10% groups were compared with middle 10% group to evaluated the variety of each radiological parameters statistically. The purpose of group comparison was described in more detail at “Analysis of radiologic parameters”.

  1. Was there really no statistical significant differences between genders? It would be useful to see that in a table or figure.

RESPONSE:

Thank you for your valuable comment. In this study, radiologic parameters were not significantly different between male and female groups. These results were additionally presented as Table 3 at “Results”

  1. Results such as "larger inter spine distance equaled lateralised spine and vise versa" refer to a more general knee anatomy and is correlated with the size of the femoral notch (stenotic,type A or W) .It would be interesting to see that in the future.

RESPONSE:

Thank you for your valuable comment. We agree that femoral notch size can be correlated with inter-spine distance, and anatomical position of lateral tibial spine, and it can be interesting issue in the future study. We described this point at “Discussion”.

  1. Last but not least serious English grammar and spelling review is required.

RESPONSE:

Thank you for your valuable comment. English grammar and spelling were reviewed again by native speaker.

Comments on the Quality of English Language

  1. Line 27 : canbeaffected

RESPONSE:

Thank you for your valuable comment. We corrected as your comment.

  1. Line 56: posterior

RESPONSE:

Thank you for your valuable comment. We corrected as your comment.

  1. Line 90 : analyze

RESPONSE:

Thank you for your valuable comment. We corrected as your comment.

  1. Line 108: KL II

RESPONSE:

Thank you for your valuable comment. We corrected as your comment.

  1. Full English gramar review please. Shorter sentences to make the reading easier.

RESPONSE:

Thank you for your valuable comment. English grammar and spelling were reviewed again by native speaker.

Please see that attachment

Reviewer 2 Report

This study entitled “Analysis of lateral tibial spine as an atomical landmark for weight bearing line assessment in high tibial osteotomy” seems to have been generally well executed and written. Furthermore, I believe that this paper will be of great interest to the readers. However, I have only a few remarks that require authors attention.

Title

Please add the type of article in your title. 

Introduction

Please state the clear hypothesis of your study at the end of Introduction.

Materials and Methods

Study population

Please add in the first sentence where the study was conducted. 

Ethical approval

This section should be a part of the first section i.e., part of Study population. Please include the number of approval and the date of approval.

Statistical Analysis

Although, you performed a retrospective study, sample size calculation should be done.

Did you apply a Bonferroni correction into your calculations? 

Results

State exact P values not just P < 0.05

Discussion

Discussion is too short. Please, expand this section with the novel findings of similar studies.

Author Response

  1. Title

Please add the type of article in your title. 

RESPONSE:

Thank you for your valuable comment. We corrected title as your comment.

  1. Introduction

Please state the clear hypothesis of your study at the end of Introduction.

RESPONSE:

Thank you for your valuable comment. We added hypothesis at the end of Introduction.

  1. Study population

Please add in the first sentence where the study was conducted. 

RESPONSE:

Thank you for your valuable comment. The location where the study was conducted was described in the first sentence of “Study population”

4.Ethical approval

This section should be a part of the first section i.e., part of Study population. Please include the number of approval and the date of approval.

RESPONSE:

Thank you for your valuable comment. Ethical approval was moved to “Study population” with the number and date of approval.                                         

  1. Statistical Analysis

Although, you performed a retrospective study, sample size calculation should be done.

Did you apply a Bonferroni correction into your calculations? 

RESPONSE:

Thank you for your valuable comment. Sample size calculation was described at “ Statistical Analysis”. In this study, two group comparisons (upper 10% vs middle 10%, and middle 10% vs lower 10%) were performed, and Bonferroni correction was not performed.

  1. Results

State exact P values not just P < 0.05

RESPONSE:

Thank you for your valuable comment. Exact p-value of each parameter was presented at the results.

  1. Discussion

Discussion is too short. Please, expand this section with the novel findings of similar studies.

RESPONSE:

Thank you for your valuable comment. We added discussion related with different results between our study and another similar study.

Round 2

Reviewer 1 Report

The authors have not answered to my question why have they not used a long leg view. Even though their calibration is accepted for the specific study,to measure specific anatomical landmarks of the tibial plateau, a proper HTO planning should always be done using a long leg view ,aquiring the Mikulicz line , mPTA, mLDFA ,JLCA and the Miniaci method for correction angle . At least, these parameters should be described in the paper , or a clear reference to the authors algorithm for HTO preop planning.Do the authors really perform their osteotomies with plain standing AP views of the Knee? 

I am happy with the rest. I appreciate the table with gender specific numbers.

Author Response

The authors have not answered to my question why have they not used a long leg view. Even though their calibration is accepted for the specific study,to measure specific anatomical landmarks of the tibial plateau, a proper HTO planning should always be done using a long leg view ,aquiring the Mikulicz line , mPTA, mLDFA ,JLCA and the Miniaci method for correction angle . At least, these parameters should be described in the paper , or a clear reference to the authors algorithm for HTO preop planning.Do the authors really perform

RESPONSE:

Thank you for your valuable comment. We absolutely agree with your comment that a long leg view is essential for pre-operative planning for HTO, and we also use a long leg view for calculation of correction amount before HTO. However, the subjects of this study were patients receiving outpatient knee OA treatment rather than patients scheduled for HTO. Therefore, a long leg view images were not available for all patients. However, in our hospital, patient’s posture is identical between standing knee AP view, and long leg view, and we believe that radiologic parameters related with lateral tibial spine may not different between standing knee AP view, and long leg view. We described this point as a limitation of this study.
